# The Role of Chorein Deficiency in Late Spermatogenesis

**DOI:** 10.3390/biomedicines12010240

**Published:** 2024-01-22

**Authors:** Kaoru Arai, Yoshiaki Nishizawa, Omi Nagata, Hitoshi Sakimoto, Natsuki Sasaki, Akira Sano, Masayuki Nakamura

**Affiliations:** Department of Psychiatry, Kagoshima University Graduate School of Medical and Dental Sciences, 8-35-1 Sakuragaoka, Kagoshima 890-8520, Japan; karai@m2.kufm.kagoshima-u.ac.jp (K.A.);

**Keywords:** chorea-acanthocytosis, *VPS13A*, chorein, ferroptosis, mitochondria, asthenozoospermia, IDH3A, TOM20

## Abstract

VPS13A, also known as chorein, whose loss of function causes chorea-acanthocytosis (ChAc), is characterized by Huntington’s-disease-like neurodegeneration and neuropsychiatric symptoms in addition to acanthocytosis in red blood cells. We previously reported that ChAc-model mice with a loss of chorein function exhibited male infertility, with asthenozoospermia and mitochondrial dysmorphology in the spermatozoa. Here, we report a novel aspect of chorein dysfunction in male fertility, particularly its role in spermatogenesis and mitochondrial integrity. An increase in anti-malondialdehyde antibody immunoreaction within the testes, predominantly observed at the advanced stages of sperm formation in chorein-deficient mice, suggests oxidative stress as a contributing factor to mitochondrial dysfunction and impaired sperm maturation. The chorein immunoreactivity in spermatids of wild-type mice accentuates its significance in sperm development. ChAc-model mice exhibit mitochondrial ultrastructural abnormalities, specifically during the late stages of sperm maturation, suggesting a critical timeframe for chorein’s action in spermiogenesis. We observed an increase in TOM20 protein levels, indicative of disrupted mitochondrial import mechanisms. The concurrent decrease in metabolic enzymes such as IDH3A, LDHC, PGK2, and ACAT1 suggests a complex chorein-mediated metabolic network that is essential for sperm vitality. Additionally, heightened separation of cytoplasmic droplets from sperm highlights the potential membrane instability in chorein-deficient spermatozoa. Metabolomic profiling further suggests a compensatory metabolic shift, with elevated glycolytic and TCA-cycle substrates. Our findings suggest that chorein is involved in anti-ferroptosis and the maturation of mitochondrial morphology in the late stages of spermatogenesis, and its deficiency leads to asthenozoospermia characterized by membrane instability, abnormal cytosolic glycolysis, abnormal mitochondrial function, and a disrupted TCA cycle. Further analyses are required to unravel the molecular mechanisms that directly link these findings and to elucidate the role of chorein in spermatogenesis as well as its broader implications.

## 1. Introduction

Chorea-acanthocytosis (ChAc) is a rare neurodegenerative disease characterized by spiculated red blood cells (acanthocytosis) and Huntington’s-disease-like neuropsychiatric symptoms. Although loss-of-function mutations of the vacuolar protein sorting 13 homolog A (*VPS13A*) gene—which produces the 360 kDa protein, also known as chorein—have been identified as causative, the detailed molecular pathology remains to be elucidated [1,2,3]. Chorein has been implicated in a variety of intracellular phenomena. These include interactions with membrane cytoskeletal proteins [4], regulation of autophagy [5,6], influences on lipid transport and mitochondrial morphology [7,8], mitochondrial trafficking [9], and plasma membrane phospholipid scrambling [10].

We established a murine ChAc model (ChAc^Del/Del^) by using gene-targeting techniques to delete exons 60–61 of *Vps13a*, which are homologous to the human disease mutation [11]. These ChAc^Del/Del^ mice showed various phenotypes depending on their strain [11]. Notably, we found male infertility in the 129S6/SvEv strain, and our previous study revealed the immobility and dysmorphic mitochondria of spermatozoa in these mice [12]. Spermatozoa derived from ChAc^Del/Del^ mice constitute suitable samples for investigating the molecular pathology of ChAc because of their readily apparent phenotypes.

In this study, we undertook a series of targeted experiments to explore specific aspects of chorein’s role in spermatogenesis. Firstly, we examined the impact of chorein deficiency on the mitochondrial structure within sperm cells during the process of spermatogenesis, utilizing transmission electron microscopy (TEM) to reveal mitochondrial dysmorphology. Secondly, we conducted a proteomic analysis of chorein co-immunoprecipitated (co-IP) proteins in murine sperm lysates to detect chorein-interacting proteins. This was followed by Western blotting and immunohistochemical analysis of murine sperm and testes based on the proteomics results. We also performed metabolomic analyses on sperm from ChAc^Del/Del^ and wild-type (WT) mice to investigate enzymes involved in energy metabolism. Additionally, building on our preliminary research with human embryonic kidney 293 (HEK293) cells, which indicated an association between *VPS13A* knockdown and ferroptosis, we specifically stained for lipid peroxidation within the testes during the process of spermatogenesis, using an anti-malondialdehyde antibody to identify increased oxidative stress, particularly during the advanced stages of spermatogenesis.

## 2. Materials and Methods

### 2.1. Animals

This study was conducted in accordance with the Guidelines for Animal Experimentation (MD14017) and Gene Recombination Experiments (24054) of the Graduate School of Medical and Dental Sciences, Kagoshima University, Japan. 

We previously generated mice harboring deletion of exons 60–61 in *Vps13a* [11]. In this study, we used an inbred strain, 129S6/SvEv (Taconic Labs, Hudson, NY, USA); 129S6/SvEv-*Vps13a^tm1asan^*/*Vps13a^tm1asan^* (ChAc^Del/Del^) was established previously [11].

### 2.2. TEM Analysis

TEM analysis was conducted as previously described [12]. Briefly, murine testes and epididymides were removed and fixed with 2% paraformaldehyde (PFA) and 2% glutaraldehyde (GA). The samples were then postfixed with 2% osmium tetroxide (OsO_4_) and embedded in resin (Quetol-812; Nisshin EM Co., Tokyo, Japan). The polymerized resins were divided into ultrathin sections and stained with 2% uranyl acetate and lead stain solution (Sigma-Aldrich Co., Tokyo, Japan). The sections were observed using a transmission electron microscope (JEM-1400Plus; JEOL Ltd., Tokyo, Japan). Images were taken with a CCD camera (EM-14830RUBY2; JEOL Ltd., Tokyo, Japan).

### 2.3. Co-Immunoprecipitation Analyses

Co-IP and reverse co-IP assays were performed using Dynabeads Protein G (Thermo Fisher Scientific, Rockford, IL, USA), according to the manufacturer’s instructions. Anti-chorein antibody, anti-IDH3A protein antibody, rabbit IgG (NB810, Novus Biologicals, Centennial, CO, USA), and goat IgG (sc2028, Santa Cruz biotechnology, Inc., Dallas, TX, USA) were used for the Dynabeads–antibody complex. Epididymal sperm from eight wild-type mice were washed with phosphate-buffered saline (PBS) and centrifuged (10,000× *g* for 10 min). Spermatozoa were solubilized in 1% NP-40 with 1× protease inhibitor cocktail (M-PER; Roche Diagnostics, Indianapolis, IN, USA) and 0.5 mM phenylmethylsulfonyl fluoride at 4 °C for 60 min. A total of 500 µg of the soluble fractions of the spermatozoa lysates (input) was incubated for 10 min at room temperature with the Dynabeads–antibody complex. Then, the beads were washed three times with lysis buffer before elution. The protein samples were eluted with 1× NuPAGE LDS Sample Buffer (Life Technologies Corporation, Carlsbad, CA, USA) and 2.5% 2-mercaptoethanol. The eluates were analyzed via proteomics and/or immunoblot analysis. Co-IP and reverse co-IP assays of proteins extracted from HEK293 cells were also performed in the same manner.

### 2.4. Proteomic Analysis

The eluates of chorein co-IP and IgG co-IP (negative control) were boiled in 60 µL of 1.5× LDS buffer for 15 min. Half of each sample was processed by SDS-PAGE using a NuPAGE 10% Bis-Tris Gel with the MES buffer system (Life Technologies Corporation, Carlsbad, CA, USA). The mobility region was excised into 10 equally sized segments, and in-gel digestion with trypsin was performed. Half of each digestate was analyzed by nano-liquid chromatography coupled with tandem mass spectrometry (LC-MS/MS), using a Waters NanoAcquity HPLC system interfaced with a Fusion Lumos mass spectrometer (Thermo Fisher Scientific, Rockford, IL, USA). Peptides were loaded on a trapping column and eluted over a 75 µm analytical column at 350 nL/min; both columns were packed with Luna C18 resin (Phenomenex, Torrance, CA, USA). The mass spectrometer was operated in data-dependent mode, with the Orbitrap operating at 70,000 FWHM and 17,500 FWHM for MS and MS/MS, respectively. The fifteen most abundant ions were selected for MS/MS. The instrument was run with a 3 s cycle for MS and MS/MS, with advanced peak determination (APD) enabled. A total of 5 h of instrument time was employed per sample. Data were sought using a local copy of Mascot (Matrix Science Inc., Boston, MA, USA). Mascot DAT files were parsed into Scaffold version 5.3.0 (Proteome Software Inc., Portland, OR, USA) for validation and filtering and to create a non-redundant list for each sample. Data were filtered using 1% protein and peptide FDR and required at least two unique peptides per protein. Protein mass spectrometry was carried out at MS Bioworks LLC., Ann Arbor, MI, USA.

The identified proteins were screened based on the spectral count (SpC). The spectral counting method is a relative quantification method comparing the SpC of each protein across LC-MS/MS runs. We selected proteins with at least SpC > 4 in the chorein co-IP samples and detected those with a fourfold or greater increase compared to the negative control sample based on dividing the SpC.

### 2.5. Metabolomic Analysis

Epididymal spermatozoa from five WT and five ChAc^Del/Del^ mice were individually analyzed for intracellular metabolites. The spermatozoa were suspended in PBS and collected by centrifugation (500× *g* and 4 °C for 5 min). The cells were then treated with 800 µL of methanol and vortexed for 30 s. Next, the cell extract was treated with 550 µL of Milli-Q water containing internal standards (H3304-1002 [13], Human Metabolome Technologies, Tsuruoka, Japan) and vortexed for 30 s. The extract was obtained and centrifuged at 2300× *g* and 4 °C for 5 min, and then, 800 µL of the upper aqueous layer was centrifugally filtered through a Millipore 5 kDa cutoff filter at 9100× *g* and 4 °C for 120 min to remove proteins. The filtrate was centrifugally concentrated and resuspended in 25 µL of Milli-Q water for capillary electrophoresis time-of-flight mass spectrometry analysis. Metabolomic analysis was carried out through a facility service at Human Metabolome Technologies. For the metabolites that were able to determine the standard deviation for both groups, the Welch’s *t*-test was performed for statistical analysis.

### 2.6. Antibodies

Antibodies against the following proteins were purchased: chorein, NBP1-85641 (Novus Biologicals, Centennial, CO, USA); phosphoglycerate kinase 2 (PGK2), 13686-1-AP (Proteintech Group, Rosemont, IL, USA); lactate dehydrogenase C (LDHC), PA5-18779 and acetyl-CoA acetyltransferase 1 (ACAT1), PA5-34676 (Thermo Fisher Scientific, Rockford, IL, USA); isocitrate dehydrogenase 3 subunit alpha (IDH3A), ab118278, annexin A1 (ANXA1), ab214486 and outer dense fiber protein 1 (ODF1), ab197029 (Abcam, Cambridge, UK); translocase of outer membrane 20 kDa subunit (TOM20), sc-11415 (Santa Cruz Biotechnologies, Dallas, TX, USA); alpha-tubulin, 11H10 (Cell Signaling Technology, Danvers, MA, USA); malondialdehyde (MDA), MA5-27559 (Thermo Fisher Scientific).

### 2.7. Western Blotting Analysis

In addition to eluates of the co-IPs, lysates of testes and epididymal sperm from five wild-type and five ChAc^Del/Del^ mice were individually solubilized with RIPA buffer (Thermo Fisher Scientific, Rockford, IL, USA) and analyzed by Western blotting. The proteins were denatured with NuPAGE LDS Sample Buffer (Life Technologies Corporation), separated on appropriate gels and transferred to polyvinylidene difluoride membranes (Merck KGaA, Darmstadt, Germany). The amount of protein sample was visualized using the MemCode Reversible Protein Stain Kit (Thermo Fisher Scientific). Membranes were blocked for 1 h at room temperature with 3% non-fat dried milk in PBS containing 0.1% Tween-20, and then, they were incubated with primary antibodies (chorein, PGK2, LDHC, ACAT1, IDH3A, ANXA1, ODF1, TOM20, and alpha-tubulin) and the appropriate secondary antibodies. Proteins were visualized using ECL Prime Western Blotting Detection Reagent (GE Healthcare, Buckinghamshire, UK), and images were recorded with a digital analyzer (FUSION-SOLO.7S.WL; Vilber-Lourmat, Marne-la-Vallée, France). ImageJ software version 1.52a (U.S. National Institutes of Health, Bethesda, ML, USA) was used for the densitometric analysis of protein bands. The results were standardized to alpha-tubulin. The Mann–Whitney U test was performed for statistical analysis using EZR in R Commander version 1.27 [14].

### 2.8. Immunohistochemistry and Immunocytochemistry

Immunohistochemical analysis was performed as previously described in [12]. Briefly, the testes were removed and fixed in 4% paraformaldehyde. The tissues were frozen and cut into 25 µm sections, which were mounted on slide glasses. Suspended spermatozoa were treated with 4% paraformaldehyde and washed in 10 mM glycine in PBS, and then, they were mounted on glass slides and air-dried. The tissues and cells were treated with the appropriate blocking buffer and incubated with primary antibodies (chorein, PGK2, LDHC, and IDH3A) and the appropriate secondary antibodies. The immunoreactions were visualized using the avidin-biotinylated enzyme complex method (VECTASTAIN ABC kit, Vector Laboratories, Burlingame, CA, USA) or the immunofluorescence technique using secondary antibodies (Alexa Fluor 555 Donkey anti-goat IgG (H + L) and/or Alexa Fluor 480 donkey anti-rabbit IgG (H + L), Invitrogen, Eugene, OR, USA). For immunofluorescence, the coverslips were washed and mounted with Vectashield medium containing DAPI (Vector Laboratories) and then viewed with a BZ-X 710 fluorescence microscope system (Keyence, Osaka, Japan).

## 3. Results

### 3.1. TEM Analysis

In this study, we utilized TEM to analyze seminiferous tubules from both wild-type and ChAc^Del/Del^ mice, aiming to identify the timing of morphological abnormalities in the sperm mitochondria during spermatogenesis in ChAc^Del/Del^ mice. Upon extensive observation of the seminiferous tubules, we detected these morphological abnormalities at a late stage of the mitochondrial sheath-development process (Figure 1). While the electron density in the mitochondrial matrix was similar in sperm from the seminiferous tubules of both wild-type and ChAc^Del/Del^ mice, significant differences were observed in sperm from the epididymis. Additionally, in the epididymal sperm of ChAc^Del/Del^ mice, we noted the destruction of cristae structures.

### 3.2. Proteomic Analysis

Co-IP was conducted to identify chorein-interacting proteins. The consecutive proteomic analyses identified 129 proteins in the chorein co-IP sample. Those proteins were screened based on SpC, and the results indicated IDH3A, PGK2, ANXA1, ODF1, GSTM5, LDHC, and ACAT1 as candidate proteins (Figure 2). To confirm the results of the proteomic analyses, chorein-co-immunoprecipitated proteins were evaluated by Western blotting. IDH3A and ACAT1 bands were detected on the immunoblotted membrane, but no bands of other co-immunoprecipitated proteins were present. On the other hand, both chorein and PGK2 bands were detected in IDH3A immunoprecipitants (Figure 2).

### 3.3. Western Blotting Analysis for Testis and Sperm Lysates

Western blotting analysis for epididymal sperm and testis lysates was performed to compare the PGK2, IDH3A, LDHC, TOM20, and ACAT1 protein immunoreaction levels of ChAc^Del/Del^ mice with those wild-type mice. The MemCode Reversible Protein Stain Kit exhibited different patterns of protein staining between WT and ChAc ^Del/Del^ mice. However, the testis lysates exhibited similar patterns in both groups (Figure 3A). The results of the immunoblotting of sperm revealed that ChAc^Del/Del^ mice exhibited significantly lower levels of PGK2 and IDH3A. The band densities of LDHC and ACAT1 in ChAc^Del/Del^ mice tended to be lower than those in WT mice. ChAc^Del/Del^ mice exhibited significantly higher levels of TOM20, i.e., the mitochondrial outer membrane protein, than WT mice. There were no differences in the protein immunoreactivity levels of testis lysates between WT and ChAc^Del/Del^ mice (Figure 3B,C).

### 3.4. Metabolomic Analysis

The relative values of metabolites were elucidated along the pathway of energy metabolism. The majority of metabolites in the glycolysis and tricarboxylic acid (TCA) cycle pathway showed higher values in ChAc^Del/Del^ mice than in WT mice. This trend was not observed for 3-phosphoglyceric acid (3PG), 2PG (2-phosphoglyceric acid), PEP (phosphoenolpyruvate), and acetyl-CoA, which are downstream of PGK2 in the glycolysis pathway (Figure 4). These results are consistent with the decreased numbers of mitochondrial enzymes in ChAc mice.

### 3.5. Immunohistochemical and Immunocytochemical Analyses

Immunohistochemical analysis revealed that the immunoreactivity and localization of chorein, IDH3A, PGK2, and LDHC proteins were similar in the testes or epididymis between WT and ChAc^Del/Del^ mice. Chorein immunoreaction was homogeneously and granularly detected in the cytoplasm of spermatids. PGK2 and LDHC were enriched through spermatogenesis in the cytoplasm of spermatids and cytoplasmic droplets of spermatozoa in the ductus epididymis. In contrast, IDH3A immunoreaction was detected in spermatogonia but not in spermatids and spermatozoa in the ductus epididymis (Figure 5A). Immunocytochemical analysis revealed that chorein immunoreaction was diffused in the midpiece of spermatozoa that were collected by the swim-up method from the epididymis, while IDH3A was localized at both ends of the midpiece (Figure 5B). IDH3A immunoreaction was visually observed in the majority of WT mice but the minority of ChAc^Del/Del^ mice (Figure 5C).

In the immunocytochemical assay of MDA in the seminiferous tubules, immunoreactions were observed predominantly in the final stages of spermatogenesis (Figure 6). In wild-type mice, the cilia were primarily lightly stained, whereas in ChAc^Del/Del^ mice, there was relatively strong staining in the region encompassing the midpiece of the sperm (magnified image in Figure 6). This suggests that ChAc^Del/Del^ mice exhibit increased membrane peroxidation in the region inclusive of the midpiece at the final stages of spermatogenesis.

## 4. Discussion

The phenotypes of ChAc^Del/Del^ mice do not completely mimic human ChAc [11]. Some of the predominant symptoms of ChAc, such as involuntary orofacial movement, dystonia, psychotic symptoms, etc., could not be clearly detected in ChAc^Del/Del^ mice. Recently, male infertility has been reported in human ChAc patients [15]. In this situation, male infertility as a clear-cut phenotype in ChAc^Del/Del^ mice would provide an ideal target to understand chorein’s function at the molecular level. We previously reported asthenozoospermia with dysmorphic mitochondria in the spermatozoa of ChAc^Del/Del^ mice [12]. However, the morphological abnormalities of the mitochondria during spermatogenesis, along with the detailed molecular pathology, remain poorly understood in ChAc^Del/Del^ mice.

In the present study, we performed TEM analysis of the seminiferous tubules in ChAc^Del/Del^ mice to detect morphological abnormalities of the mitochondria during spermatogenesis. We observed distinctive mitochondrial dysmorphology in the late stages of the mitochondrial sheath development process in ChAc^Del/Del^ mice, as illustrated in Figure 1. Similarities in electron density within the mitochondrial matrix of sperm from the seminiferous tubules of both wild-type and ChAc^Del/Del^ mice suggest that chorein deficiency may not significantly alter the basic mitochondrial structure during the early stages of spermatogenesis. However, the notable differences observed in sperm from the epididymides of ChAc^Del/Del^ mice—particularly the destruction of cristae structures—indicate a progressive deterioration of mitochondrial integrity in the absence of chorein. Additionally, chorein immunoreaction was homogeneously and granularly detected in the cytoplasm of spermatids (Figure 5A), aligning with and underscoring the critical role of chorein in maintaining mitochondrial structure and function during the later stages of sperm development. These observations raise intriguing questions about the specific mechanisms by which chorein deficiency leads to mitochondrial dysmorphology and subsequent sperm dysfunction. The progressive nature of mitochondrial deterioration observed during spermatogenesis suggests that chorein may play a more crucial role in the later stages of spermatogenesis, particularly in the maturation and maintenance of sperm cells.

Co-IP and subsequent proteomic analyses were conducted to identify chorein-interacting proteins in mouse sperm. The results and reverse co-IP revealed chorein–IDH3A interaction (Figure 2). The IDH3A protein is the alpha subunit of NAD (+)-dependent isocitrate dehydrogenase, which is a TCA-cycle enzyme localized in the mitochondria. It was shown that oxidative stress induced by 6-hydroxydopamine (6OHDA) or rotenone can alter *IDH3A* and high-temperature-requirement protein A2 (*HTRA2/OMI*) (a mitochondrial protease) gene expression [16]. A recent study on flies reported that *idh3a* knockout caused dysmorphic mitochondria that lacked cristae in *idh3a*-knockout flies [17]. These studies suggest that IDH3A may play a crucial role in mitochondrial quality control. Disorganized inner membrane structures of mitochondria were observed in the sperm of ChAc^Del/Del^ mice [12]. However, to elucidate the details of the relationships between chorein, IDH3A, and dysmorphic mitochondria, further investigations are needed. Another study showed retinal degeneration and mitochondrial dysfunction in *Idh3a*-mutant mice [18], but no other detectable abnormal phenotypes were observed. The sperm mobility and morphology of *Idh3a*-mutant mice remain unknown.

Immunocytochemical analysis revealed that IDH3A was ubiquitously distributed in spermatogonia and disappeared in spermatids or spermatozoa packed in the ductus epididymis (Figure 5A). IDH3A immunoreaction returned in the spermatozoa and was localized at both ends of the midpiece (Figure 5B) in WT mice. However, there was no immunoreaction of IDH3A in the majority of ChAc^Del/Del^ spermatozoa (Figure 3C), which we previously reported as immotile. These findings suggest that IDH3A may be involved in murine sperm motility.

In our study, we observed that protein staining in sperm exhibited different patterns between WT and ChAc^Del/Del^ mice (Figure 3A), suggesting alterations in protein expression or localization during spermatogenesis. This is particularly notable because testis lysates from both groups displayed similar staining patterns. The disparity in sperm protein staining patterns, in contrast to the uniformity observed in testis lysates, indicates specific disruptions in the later stages of spermatogenesis in ChAc^Del/Del^ mice. These disruptions could be associated with the morphological abnormalities in mitochondria identified during these stages by TEM analysis. In addition, the Western blotting analysis revealed a decrease in band density for IDH3A and ACAT1, whereas an increase was noted for TOM20 in ChAc^Del/Del^ mice (Figure 3B). This contrast in protein levels—particularly the increase in TOM20, which is a mitochondrial outer membrane protein—suggests a compensatory response to the mitochondrial dysmorphology observed in these mice. Notably, such differences were not observed in the testis lysates of either group, indicating that these changes likely occur during the later stages of spermatogenesis. The alterations in protein levels, especially the increase in TOM20 in sperm, are consistent with the morphological abnormalities in mitochondria detected during the later stages of spermatogenesis in ChAc^Del/Del^ mice, as revealed by TEM analysis. The decrease in IDH3A and ACAT1 could be indicative of impaired mitochondrial function, while the increase in TOM20 might reflect an attempt by the cell to maintain mitochondrial integrity. This is particularly relevant given the role of TOM20 in protein import into the mitochondria, suggesting a potential adaptive mechanism to counteract mitochondrial damage. The disparity in protein expression patterns in sperm, as compared to the testes, underscores the specificity of these molecular changes to the spermatogenic process, particularly in the maturation phase of spermatozoa. Given the crucial role of mitochondria in energy production and cellular metabolism within sperm, such alterations could have significant implications for sperm motility and functionality. This is supported by the previously reported poor motility in ChAc^Del/Del^ spermatozoa [12] and provides a molecular basis for the observed sperm dysfunction.

Co-IP analysis suggested that IDH3A interacts with PGK2 (Figure 2), which is a glycolytic enzyme specifically expressed in sperm. Regarding the chorein–PGK2 interaction, the results of the proteomic analyses and Western blotting were inconsistent. This suggests that both chorein and PGK2 interact with IDH3A but that chorein and PGK2 may interact indirectly.

PGK2 and LDHC, another sperm-specific glycolytic enzyme, are known as essential proteins for sperm motility and male fertility in mice [19,20]. The reductions in those enzymes observed in the sperm of ChAc^Del/Del^ mice were consistent with sperm immobility. Although the results of immunohistochemical analysis and Western blotting of testis lysates suggested that the PGK2 and LDHC proteins were expressed in similar patterns in the sperm of both WT and ChAc^Del/Del^ mice (Figure 5A), Western blotting of sperm lysates after washing in PBS and centrifugation showed reduced immunoreaction of those proteins in ChAc^Del/Del^ mice relative to WT mice (Figure 2). PGK2 and LDHC were enriched in the cytoplasmic droplet, a cytoplasm-containing structure attached to epididymal sperm. Yuan et al. reported that cytoplasmic droplets play essential roles in sperm maturation, including mitochondrial activation in the epididymis [21]. They also showed that the function of the cytoplasmic droplet is abrogated by removing it from sperm through mechanical stimulation. We thus hypothesized that the sperm of ChAc^Del/Del^ mice might have a fragile plasma membrane and that their cytoplasmic droplets might be easily removed by mechanical stimulation, consequently leading to impaired sperm motility.

Building on our preliminary research in human embryonic kidney 293 (HEK293) cells, which indicated an association between *VPS13A* knockdown and ferroptosis, in the present study, we performed immunocytochemistry of MDA to stain for lipid peroxidation within the testes during spermatogenesis. Interestingly, in the final stage of spermatogenesis, spermatozoa of ChAc^Del/Del^ mice showed increased staining in the region encompassing the midpiece of the sperm (Figure 6). This suggests that ferroptosis may occur in ChAc^Del/Del^ mice during the final stages of spermatogenesis. Activating lipid peroxidation through ferroptosis possibly causes fragile mitochondrial dysmorphological changes and plasma membrane changes in the late stages of spermatogenesis. These may explain the previously described abnormalities observed in the late stages of the spermatogenesis of ChAc^Del/Del^ mice.

Metabolomic analyses detected widely increased metabolites of the glycolysis and TCA cycles in ChAc^Del/Del^ mice relative to WT mice (Figure 4), aligning with the commonly reported metabolomic profiles in mitochondrial disease patients and animal models [22]. In contrast, a metabolomic study using human sperm cells from idiopathic asthenozoospermia patients showed a decrease in these metabolites [23]. It was also reported that *Idh3a*-knockout flies showed decreased levels of TCA-cycle substrates [17]. Concerning energy metabolism, the results should be interpreted with caution because the samples were epididymal spermatozoa. Since mitochondria are inactive in epididymal spermatozoa, glycolytic enzymes that are abundant in the cytoplasmic droplet have been shown to play an important role in ATP production [21]. Spermatozoa from ChAc^Del/Del^ mice have intact cytoplasmic droplets when stored in the epididymis, suggesting that they are capable of ATP production. This is consistent with the previous study that showed there is no difference in the amount of ATP in epididymal spermatozoa of WT and ChAc^Del/Del^ mice [12]. However, as previously mentioned in the discussion, there is a potential vulnerability of cytoplasmic droplets in ChAc^Del/Del^ mouse sperm to mechanical stimulation. Additionally, the impact of these metabolic changes on sperm function in ChAc^Del/Del^ mice has not been fully elucidated. These require further investigation into the mechanisms underlying sperm dysfunction in this model.

Because we focused on glycolytic enzymes and TCA-cycle enzymes in the present study, we did not expend substantial effort on ACAT1, ANXA1, GSTM5, and ODF1. Chorein–ACAT1 interaction was suggested by chorein co-IP analysis (Figure 2). ACAT1, a protein localized in the mitochondria, plays a pivotal role in fatty acid metabolism and energy production. This enzyme catalyzes the reversible formation of acetoacetyl-CoA from two molecules of acetyl-CoA. ACAT1’s involvement extends to the processing of acetyl-CoA produced during beta-oxidation, a critical pathway for energy production. Furthermore, the acetoacetyl-CoA generated by ACAT1 is utilized in the TCA cycle, which is essential for cellular energy supply. The reduction in ACAT1 in the sperm of ChAc^Del/Del^ mice, as indicated by our findings, may contribute to a complex metabolic scenario. While ACAT1 impairment primarily impacts fatty acid metabolism, this could indirectly influence the TCA cycle. Decreased beta-oxidation activity might necessitate enhanced glycolysis and TCA-cycle activity to meet energy demands, possibly explaining the increased levels of TCA-cycle metabolites observed. This interplay illustrates the intricate balance of cellular metabolic pathways and highlights the need for an integrated understanding of how various metabolic alterations, including those involving ACAT1, impact overall cellular function, especially in the context of chorein-deficient sperm. Future studies delving deeper into these metabolic networks will be essential to unravel the detailed pathological changes and their broader implications.

In conclusion, our findings highlight the critical role of chorein in maintaining mitochondrial integrity during sperm development and underscore the complexities of mitochondrial regulation in spermatogenesis. The observed protein expression changes in ChAc^Del/Del^ mice, particularly the increase in TOM20 and the decreases in IDH3A and ACAT1, offer insights into the molecular mechanisms underlying the mitochondrial dysmorphology and asthenozoospermia in these mice. Furthermore, the increased lipid peroxidation and staining patterns that we observed suggest the involvement of ferroptosis, particularly in the advanced stages of spermatogenesis. This potential association between ferroptosis and the observed mitochondrial and cellular abnormalities in ChAc^Del/Del^ mice provides a new perspective on the molecular pathology of ChAc underlying these defects and opens up avenues for further research into the role of ferroptosis in spermatogenesis and male infertility. Further studies are required to elucidate the precise molecular pathways through which chorein deficiency leads to ferroptosis and its impact on sperm development. This exploration could potentially expand our understanding of male reproductive health.

## Figures and Tables

**Figure 1 biomedicines-12-00240-f001:**
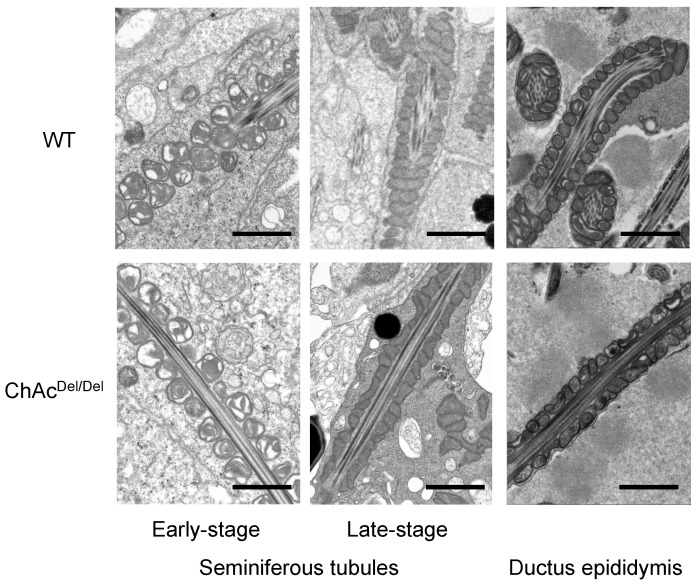
Transmission electron microscopy of mitochondria during spermatogenesis in wild-type (WT) and ChAc^Del/Del^ mice. Left column: Early-stage spermatids in the seminiferous tubule, with no discernible differences between the mitochondria of WT and ChAc^Del/Del^ mice. Middle column: Late-stage spermatids in the seminiferous tubule, where the mitochondria of ChAc^Del/Del^ mice exhibit distorted cross-sections. Right column: Epididymal spermatozoa, showing morphological distortions and changes in electron density within the mitochondria of ChAc^Del/Del^ mice. Scale bars = 1 µm.

**Figure 2 biomedicines-12-00240-f002:**
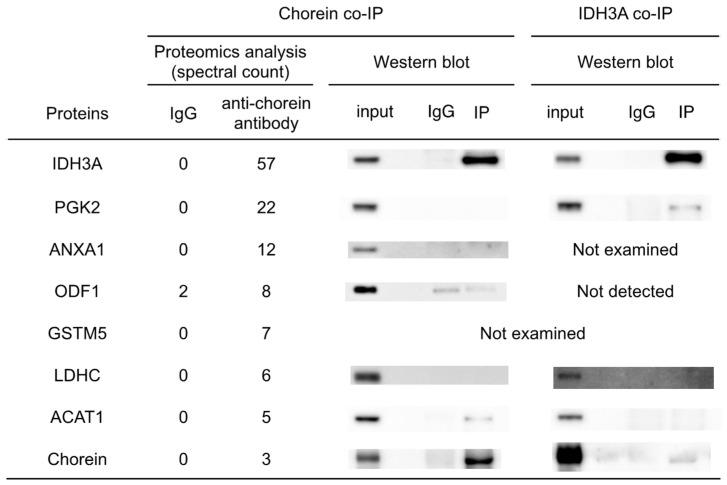
Results of immunoprecipitation analysis. Seven proteins were identified as candidate chorein-interacting proteins by proteomic analysis based on the spectral counting method. IDH3A and ACAT1 were detected by Western blotting analysis following chorein co-IP. Chorein and PGK2 were detected by Western blotting analysis following IDH3A co-IP.

**Figure 3 biomedicines-12-00240-f003:**
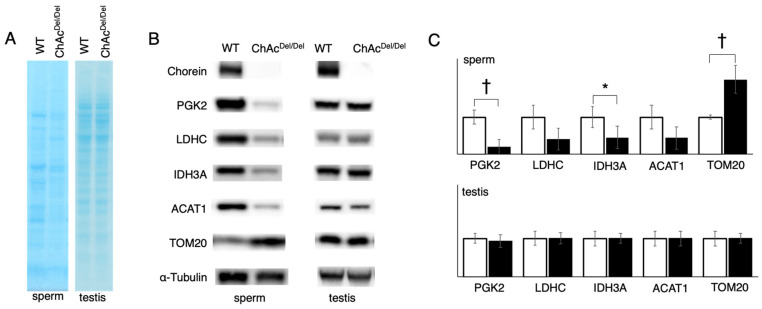
Western blotting analysis for epididymal sperm and testis lysates was performed to compare the protein levels of ChAc^Del/Del^ mice (n = 5) with those of WT mice (n = 5). (**A**) Proteins were visualized using the MemCode Reversible Protein Stain Kit after separation by SDS-PAGE and transfer to PVDF membranes. The contrast of the sperm lysates image was increased by a factor of two to enhance visibility. Sperm lysates exhibited different patterns of protein expression between WT and ChAc^Del/Del^ mice. Testis lysates exhibited similar patterns between both groups. (**B**) Representative results of Western blotting analysis are shown. (**C**) Relative quantification of Western blotting analysis is presented with white and black bars indicating WT and ChAc mice, respectively. The Mann–Whitney U test was performed for statistical analysis. WT sperm lysates exhibited significantly higher levels of IDH3A and PGK2 than ChAc^Del/Del^ sperm lysates. Interestingly, WT mice exhibited significantly lower levels of TOM20 than ChAc^Del/Del^ mice. No differences were observed between WT and ChAc^Del/Del^ testis lysates. * *p* < 0.05; ^†^ *p* < 0.01.

**Figure 4 biomedicines-12-00240-f004:**
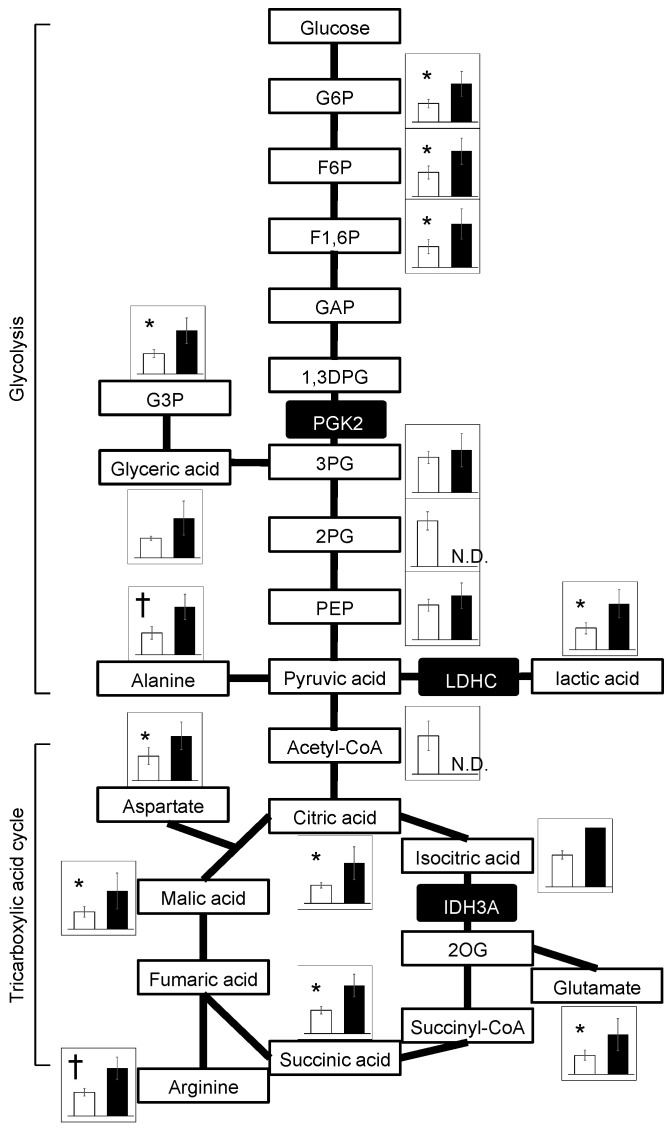
Metabolites of the glycolysis and TCA-cycle pathways obtained from metabolomic analysis. Bar graphs represent the relative amounts of metabolites. White and black bars indicate WT mice (n = 5) and ChAc mice (n = 5), respectively. White boxes represent metabolites. Black boxes represent enzymes. PGK2, LDHC, and IDH3A are enzymes detected by chorein co-IP and proteomic analysis of sperm lysates; Welch’s *t*-test was performed. * *p* < 0.05; ^†^ *p* < 0.01; N.D., not detected. Abbreviations; 1,3DPG, 1,3-bisphosphoglycerate; 2OG, 2-oxoglutarate; 2PG, 2-phosphoglyceric acid; 3PG, 3-phosphoglyceric acid; G3P, glyceraldehyde 3-phosphate; G6P, glucose-6-phosphate; GAP, glyceraldehyde 3-phosphote; F1,6P, fructose 1,6-bisphosphate; F6P, fructose 6-phosphate; PEP, phosphoenolpyruvate.

**Figure 5 biomedicines-12-00240-f005:**
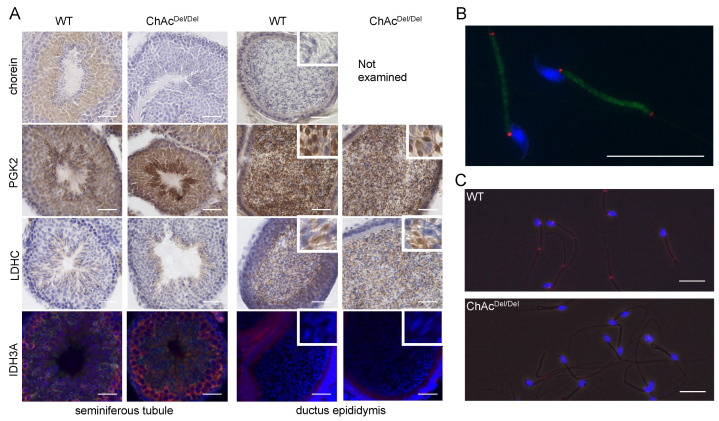
(**A**) Localization of chorein, PGK2, LDHC, and IDH3A proteins in the seminiferous tubule/ductus epididymis. Small windows show magnified images. Chorein, PGK2, and LDHC were each stained with appropriate antibodies (brown). The nuclei were stained with hematoxylin (blue). Cytoplasm in elongated spermatids and cytoplasmic droplets in epididymal sperm were stained with antibodies to the PGK2 protein. The pattern of LDHC protein immunoreaction was consistent with that of PGK2. Immunolabeled IDH3A protein (red) was observed in spermatogonia but not in spermatids or epididymal sperm. The acrosome sacs and nuclei were stained with PNA (green) and DAPI (blue), respectively. Scale bars = 50 µm. (**B**) Localization of chorein and IDH3A in epididymal sperm. Chorein (green) was observed in the midpiece, and IDH3A (red) was observed at both ends of the midpiece. The nuclei were stained with DAPI (blue). Scale bars = 10 µm. (**C**) The majority of the sperm of ChAc mice showed no immunoreactivity of the IDH3A protein. Scale bars = 10 µm.

**Figure 6 biomedicines-12-00240-f006:**
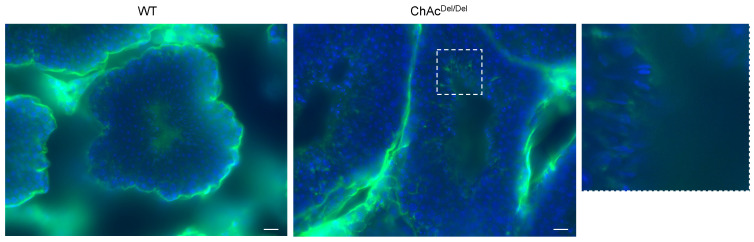
Immunohistochemical assay of malondialdehyde (MDA) in the seminiferous tubules. MDA immunoreactions (green) were observed predominantly in the final stages of spermatogenesis in both wild-type (WT) and ChAc^Del/Del^ mice. The magnified image of the region enclosed by the dashed lines is shown on the left side. In WT mice, the cilia were primarily lightly stained, whereas in ChAc^Del/Del^ mice, there was relatively strong staining in the region encompassing the midpiece of the sperm. The nuclei were stained with DAPI (blue). Scale bars = 20 µm.

## Data Availability

Data are contained within the article.

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
