# Peer review of "The Role of Chorein Deficiency in Late Spermatogenesis"

_biomedicines, 2024, doi:10.3390/biomedicines12010240_

Round 1

Reviewer 1 Report

Comments and Suggestions for Authors

In this manuscript, the Authors produced a considerable amount of data to support the view that chorein is important for the late stages of spermatogenesis.

Overall, the study was carefully conducted: the Authors performed several different analyses to investigate this topic comprehensively and the results are convincing.

 Here, a few issues that need to be addressed

 Materials & Methods.

The description of how statistical analysis was performed is lacking. In addition, information about this issue should be provided in the Figure legends, indicating the number of experiments, etc.

The conclusion that ferroptosis plays a role of in the abnormalities of spermatogenesis caused by the lack of chorein is based only on increased immunohistochemical reactivity for MDA; as the Authors acknowledged in the Discussion (lines 453-455), it is just an indication, which in my opinion does not justify the mention of ferroptosis in the title. The presence of ferroptosis should be supported by more data, for example demonstrating a role for iron.  

The Authors should check the manuscript for minor style, spelling and typographical errors.

Comments on the Quality of English Language

minor editing

Author Response

We wish to express our appreciation to the editor and reviewers for their insightful comments, which have helped us significantly improve our paper.

COMMENT #1:

In this manuscript, the Authors produced a considerable amount of data to support the view that chorein is important for the late stages of spermatogenesis.

Overall, the study was carefully conducted: the Authors performed several different analyses to investigate this topic comprehensively and the results are convincing.

RESPONSE:

Thank you very much for this insightful comment.

COMMENT #2:

The description of how statistical analysis was performed is lacking. In addition, information about this issue should be provided in the Figure legends, indicating the number of experiments, etc.

RESPONSE:

We appreciate this suggestion. We have added the following sentence: (p.10, ll.13-15)

“For the metabolites that were able to determine the standard deviation for both groups, the Welch’s t-test was performed for statistical analysis.”

We have also added information about the number of experimental replications in the figure legends. (Figure 3, Figure4)

Statistical methods for Western blotting analysis have been described at the end of Materials and Methods 7.

COMMENT #3:

The conclusion that ferroptosis plays a role of in the abnormalities of spermatogenesis caused by the lack of chorein is based only on increased immunohistochemical reactivity for MDA; as the Authors acknowledged in the Discussion (lines 453-455), it is just an indication, which in my opinion does not justify the mention of ferroptosis in the title. The presence of ferroptosis should be supported by more data, for example demonstrating a role for iron. 

RESPONSE:

We appreciate this important suggestion. We have changed the title from “Chorein Deficiency in Late Spermatogenesis: An Association between Possible Ferroptosis and Mitochondrial Dysfunction” to “The Role of Chorein Deficiency in Late Spermatogenesis”

COMMENT #4:

The Authors should check the manuscript for minor style, spelling and typographical errors.

RESPONSE:

Thank you for this comment. We have checked and revised the manuscript for minor errors.

Reviewer 2 Report

Comments and Suggestions for Authors

The study presented by Arai K. and coworkers is an attempt to solve the important physiological problem of chorein disfunction which is associated with  chorea-acanthocytosis (ChAc) and also male infertility in the mice ChAcDel/Del model which is a real advance of the authors because allowes to study the mechanism.  But the whole impression from the article is a contradictory. TEM analysis and Western blotting of mitochondria in testis and epididymal sperm lysates provide fairly convincing evidence that chorein disfunction  causes serious impairment in mitochondrial function, leading to disruption of spermatogenesis in the late stages. At the same much more data is needed to be obtained to link mitochondrial dysfunction with ferroptosis  especially if this is in the title of the article. So the title  “The Role of Chorein Deficiency in Late Spermatogenesis” describes fully the results obtained while  association with possible ferroptosis can be mentioned as indication of the useful way for proceeding of the study.

The authors compared metabolonic profiles of WT and ChAcDel/Del and found the increased level of metabolites associated with TCA-cycle and glycolysis  in ChAc mice indicating significant shift in the energetic profile of the mutated mice. At the same time  Western blots of the sperm of these mice (Fig.3) demonstrated low level of the glycolytic enzymes which is not commented in Discussion. May simply the difference in the ATP-level in WT and ChAc mice would better evaluate differences in energetics between these groups.

Technical shortcomings. Fig.3A panel must be better visualized or removed because electrophoretic bands are not seen. Fig.3C  - relative quantification of Western blots WT and ChAc mice must be identified as white and black bars like it was done on Fig.4 . Fig.4 the notes “ND” should be identified (may be as “non detected”?).

Author Response

We wish to express our appreciation to the editor and reviewers for their insightful comments, which have helped us significantly improve our paper.  

COMMENT #1:

The study presented by Arai K. and coworkers is an attempt to solve the important physiological problem of chorein disfunction which is associated with  chorea-acanthocytosis (ChAc) and also male infertility in the mice ChAcDel/Del model which is a real advance of the authors because allowes to study the mechanism.  But the whole impression from the article is a contradictory. TEM analysis and Western blotting of mitochondria in testis and epididymal sperm lysates provide fairly convincing evidence that chorein disfunction  causes serious impairment in mitochondrial function, leading to disruption of spermatogenesis in the late stages. At the same much more data is needed to be obtained to link mitochondrial dysfunction with ferroptosis  especially if this is in the title of the article. So the title  “The Role of Chorein Deficiency in Late Spermatogenesis” describes fully the results obtained while  association with possible ferroptosis can be mentioned as indication of the useful way for proceeding of the study.

RESPONSE:

We appreciate this suggestion. In accordance with your suggestion, we have changed the title from “Chorein Deficiency in Late Spermatogenesis: An Association between Possible Ferroptosis and Mitochondrial Dysfunction” to “The Role of Chorein Deficiency in Late Spermatogenesis”

COMMENT #2:

The authors compared metabolonic profiles of WT and ChAcDel/Del and found the increased level of metabolites associated with TCA-cycle and glycolysis  in ChAc mice indicating significant shift in the energetic profile of the mutated mice. At the same time Western blots of the sperm of these mice (Fig.3) demonstrated low level of the glycolytic enzymes which is not commented in Discussion. May simply the difference in the ATP-level in WT and ChAc mice would better evaluate differences in energetics between these groups.

RESPONSE:

Thank you for pointing out the need for consideration here. To address this issue, we have changed the sentence from

“Metabolomic analyses detected widely increased metabolites of the glycolytic and TCA cycles in ChAcDel/Del mice relative to WT mice (Figure 4). These findings are contrary to those of a previous study that reported decreased glycolytic and TCA-cycle metabolites in human sperm cells from idiopathic asthenozoospermia patients (23). It was also reported that idh3a-knockout flies showed decreased levels of TCA-cycle substrates (18). On the other hand, the results in this study showed a reduction in the 3-phosphoglyceric acid/fructose 1,6-bisphosphate (3PG/F1,6P) ratio in ChAcDel/Del mice relative to WT mice. This suggests a decrease in PGK2 enzyme activity and is consistent with the results of the Western blotting analysis in sperm lysates, although detailed pathological changes in the energy metabolism of the sperm of ChAcDel/Del mice require further investigation.” to

“Metabolomic analyses detected widely increased metabolites of the glycolysis and TCA cycles in ChAcDel/Del mice relative to WT mice (Figure 4), aligning with the commonly reported metabolomic profiles in mitochondrial disease patients and animal models (21). In contrast, a metabolomic study using human sperm cells from idiopathic asthenozoospermia patients showed a decrease in these metabolites (22). It was also reported that idh3a-knockout flies showed decreased levels of TCA-cycle substrates (16). Concerning energy metabolism, the results should be interpreted with caution because the samples were epididymal spermatozoa. Since mitochondria are inactive in epididymal spermatozoa, glycolytic enzymes which are abundant in the cytoplasmic droplet have been shown to play an important role in ATP production (20). Spermatozoa from ChAcDel/Del mice have intact cytoplasmic droplet when stored in the epididymis, suggesting that they are capable of ATP production. This is consistent with the previous study showed that there is no difference in the amount of ATP in epididymal spermatozoa of WT and ChAcDel/Del mice (12). However, as previously mentioned in the discussion, there is a potential vulnerability of cytoplasmic droplets in ChAcDel/Del mouse sperm to mechanical stimulation. Additionally, the impact of these metabolic changes on sperm function in ChAcDel/Del mice has not been fully elucidated. These require further investigation into the mechanisms underlying sperm dysfunction in this model.” (pp.28-29, ll.12-9)

COMMENT #3:

Technical shortcomings. Fig.3A panel must be better visualized or removed because electrophoretic bands are not seen. Fig.3C  - relative quantification of Western blots WT and ChAc mice must be identified as white and black bars like it was done on Fig.4 . Fig.4 the notes “ND” should be identified (may be as “non detected”?).

RESPONSE:

According to this suggestion, we have replaced Fig.3A panel to improve visibility by increasing the contrast of the image by a factor of two. We added the following sentence in Fig.3A legend: The contrast of the sperm lysates image has been increased by a factor of two to enhance visibility.

Also, we have added the following sentence in Fig.3C legend: White and black bars indicate WT and ChAc mice, respectively.

Concerning the description for “N.D.”, the following have been noted: “N.D., not detected” in Fig.4 legend.

Reviewer 3 Report

Comments and Suggestions for Authors

The authors of the manuscript "Chorein deficiency in late spermatogenesis:....." K. Arai et al contributed a scientifically sound paper, with presentation quality, originality and content significance for those interested in spermatogenesis and neuronal degeneration. Furthermore they offer a link between the novel cell-death mechanism of ferroptosis and spermatogenesis as this cell-death process has been already involved in cancer an mitochondrial dysfunction. In conclusion the paper presents novel experimental results and is worth publishing in "Biomedicines".   

Author Response

We wish to express our appreciation to the editor and reviewers for their insightful comments, which have helped us significantly improve our paper.

COMMENT #1:

The authors of the manuscript "Chorein deficiency in late spermatogenesis:....." K. Arai et al contributed a scientifically sound paper, with presentation quality, originality and content significance for those interested in spermatogenesis and neuronal degeneration. Furthermore they offer a link between the novel cell-death mechanism of ferroptosis and spermatogenesis as this cell-death process has been already involved in cancer an mitochondrial dysfunction. In conclusion the paper presents novel experimental results and is worth publishing in "Biomedicines".

RESPONSE:

Thank you very much for your positive feedback on our manuscript.

We are grateful for your recognition of the scientific soundness and the potential impact of our work in the field of spermatogenesis and neuronal degeneration.

Round 2

Reviewer 2 Report

Comments and Suggestions for Authors

The authors improved the quality of the Figures presented. They changed the title according to my reccomendations and now this title much better corresponds to the results obtained in the study. Besides an important paragraph was included to Discussion.